# Preparation and Characterization of D-Carvone-Doped Chitosan–Gelatin Bifunctional (Antioxidant and Antibacterial Properties) Film and Its Application in Xinjiang Ramen

**DOI:** 10.3390/foods14152645

**Published:** 2025-07-28

**Authors:** Cong Zhang, Kai Jiang, Yilin Lin, Rui Cui, Hong Wu

**Affiliations:** School of Food Science and Engineering, South China University of Technology, Guangzhou 510640, China; congz409@163.com (C.Z.); jjking2101@163.com (K.J.); echoyilin@126.com (Y.L.); cuirui910@163.com (R.C.)

**Keywords:** D-carvone, active food packaging, Xinjiang ramen preservation, composite film

## Abstract

In this study, a composite film with dual antioxidant and antibacterial properties was prepared by combining 2% chitosan and 7% gelatin (2:1, w:w), with D-carvone (0–4%) as the primary active component. The effect of D-carvone content on the performance of the composite films was systematically investigated. The results showed that adding 1% D-carvone increased the water contact angle by 28%, increased the elongation at break by 35%, and decreased the WVTR by 18%. FTIR and SEM confirmed that ≤2% D-carvone uniformly bonded with the substrate through hydrogen bonds, and the film was dense and non-porous. In addition, the DPPH scavenging rate of the 1–2% D-carvone composite film increased to about 30–40%, and the ABTS^+^ scavenging rate increased to about 35–40%; the antibacterial effect on Escherichia coli and Staphylococcus aureus increased by more than 70%. However, when the addition amount was too high (exceeding 2%), the composite film became agglomerated, microporous, and phase-separated, affecting the film performance, and due to its own taste, it reduced the sensory quality of the noodles. Comprehensively, the composites showed better performance when the content of D-carvone was 1–2% and also the best effect for freshness preservation in Xinjiang ramen. This study provides a broad application prospect for natural terpene compound-based composite films in the field of high-moisture, multi-fat food preservation, and provides a theoretical basis and practical guidance for the development of efficient and safe food packaging materials. In the future, the composite film can be further optimized, and the effect of flavor can be further explored to meet the needs of different food preservation methods.

## 1. Introduction

With growing global concern over plastic pollution, the development of biodegradable and functionalized bio-based packaging materials has become a key focus in the food industry. Chitosan (CS) and gelatin (GEL), as natural and renewable resources, have attracted much attention for their excellent biocompatibility, film-forming abilities, and antibacterial properties [1]. However, the performance limitations of individual biopolymers (e.g., brittleness and poor barrier properties of CS [2] and the high hydrophilicity and poor mechanical properties of GEL [3]) as well as safety concerns related to traditional chemical preservatives have prompted researchers to construct a new generation of smart preservation systems through synergistic integration of functionalized composites with natural active ingredients.

In recent years, the CS–GEL composite system has become a research hotspot due to its unique complementary properties [4,5]. The cationic properties of CS can penetrate microbial cell membranes and exhibit broad-spectrum antibacterial activity against both Gram-positive and Gram-negative bacteria, and its intrinsic cationic properties also provide good compatibility. In addition, under an acidic environment, the cations can produce electrostatic interactions and hydrogen bonding with the negatively charged gelatin, which makes the composites show strong mechanical and barrier properties [2,6,7]; and the high hydrophilicity, flexibility, and film-forming ability of GEL can precisely form a complementary relationship with CS, which can effectively improve the mechanical properties of the composites [7]. However, the simple CS–GEL composite system is still deficient in terms of antioxidant performance and long-lasting antibacterial efficiency. To solve this problem, researchers have explored the incorporation of plant extracts or essential oils with antioxidant and antibacterial properties into the composite system [8].

Bonilla & Sobral [9] showed that by adding cinnamon, guarana, rosemary, and boldo-do-chile ethanolic extracts to the CS–GEL film matrix not only showed good antioxidant properties and antibacterial effects but also improved the overall properties of the films. This is since these extracts are rich in polyphenols and flavonoids, which have a wide range of bioactivities, effective scavenging of free radicals, good antioxidant and anti-inflammatory properties, in addition to better biocompatibility [10]. However, their thermal instability, moisture resistance, rapid release, and migration during storage can significantly reduce their protective effect on the food storage process, making the utilization rate low [11]. For plant essential oils, which are also a hot topic in food science research nowadays, Zhao et al. [4] prepared microcapsules enriched with green tea essential oils and loaded them into GEL–CS films for strawberry preservation, which showed excellent antioxidant qualities, 60% scavenging activity against DPPH (2,2-Diphenyl-1-picrylhydrazyl), as well as good freshness retention. However, given the inherent shortcomings of plant extracts and essential oils in terms of thermal stability, volatility, and migration control, there is an urgent need for an alternative active agent that combines high stability, dual functionality, and polymer compatibility [12]. Carvone, with its ketone group on the terpenoid ring and allyl side chain, can form hydrogen bonds and hydrophobic interactions with the polysaccharide network, while providing phenolic antioxidant capacity and film-penetrating antibacterial activity, making it an ideal candidate for the construction of dual-function composite films [13].

Carvone, as a monoterpenoid, is widely found in the essential oils of plants such as peppermint (*Mentha spicata*), thyme (*Thymus vulgaris*), and wild bergamot (*Bergamia Loise var. Citrus aurantium*) [14,15]. It not only possesses strong antioxidant capacity but also disrupts microbial cell membrane integrity, and its antibacterial activity has been demonstrated in plant pathogens and food spoilage bacteria [14]. Two enantiomeric forms of carvone (L- and D-) have been identified, which have different pharmacological values. Among them, the enantiomeric form of D-carvone has shown a wide range of pharmacological effects such as antitumor, chemopreventive, antihypertensive, and antihyperlipidemic properties. Currently, D-anethole has been widely used in cancer treatment, as it can induce apoptosis, inhibit cancer cell proliferation, and suppress the growth of HL-60 cells while inducing their cell death; it also plays a role in inhibiting certain carcinogenic pathways and oncogenes (such as inhibition of cell proliferation pathways (phosphatidylinositol 3-kinase and other cell proliferation pathway markers) and the expression of c-fos and c-myc oncogenes). In addition, it can enhance the activity of antioxidants such as glutathione (GSH) and glutathione reductase (GR), and it can also catalyze the binding of GSH to various oxidants by inducing glutathione S-transferase (GST), which helps to scavenge oxidative stress products [13,16,17,18,19]. It is also used in the food industry as a preservative in food products. Ye et al. [20] found that two isomers of carvone were able to inhibit the survival of *P. aeruginosa* through oxidative damage, leading to increased protein degradation and inhibition of transcriptional levels, as well as modulating the physiological metabolic activity of *P. aeruginosa*, and generating a certain degree of drug resistance. Moreover, the substance demonstrated noteworthy antibacterial properties, exhibiting substantial efficacy against a broad spectrum of both Gram-positive and Gram-negative bacteria as well as fungi [20]. D-carvone has strong antioxidant properties, broad-spectrum antibacterial activity, and good stability, and it is green and safe. In addition, when added in small amounts (<5%), the active groups in its molecules (such as hydroxyl and ketone groups) can interact with the biopolymer network through hydrogen bonds, making it promising for use in composite materials for food packaging to simultaneously improve the mechanical properties and stability of the film. However, most existing studies are limited to performance optimization at the “material-active agent” level, and there is still a lack of “scenario–demand” coupling verification for typical perishable foods. In particular, for Chinese-style fresh wet noodle products with high moisture content, high fat content, and cold chain breakpoints, there are still no systematic reports on how to replace traditional polyethylene (PE) packaging with edible active films to achieve synergistic preservation of antibacterial, antioxidant, and water-resistant properties.

Therefore, in this study, D-carvone was used as a functional component, and it was compounded with chitosan/gelatin to construct a multifunctional active packaging film. This study analyzes the intermolecular interactions between D-carvone and CS–GEL and their effects on membrane structure and properties. It also evaluates the synergistic effects of antioxidant and antibacterial functions through DPPH/ABTS^+^ radical scavenging experiments and antibacterial zone measurements. Furthermore, using Xinjiang ramen as a model food, the inhibitory effect of the composite film on microbial growth is assessed and compared with traditional polyethylene (PE) film. Preservation efficiency of the composite film is verified in a real food system by measuring total colony count, moisture retention rate, and sensory quality). These investigations aim to provide a theoretical basis for the industrial application of natural active packaging materials.

## 2. Experimental

### 2.1. Materials

Chitosan (CS, deacetylation degree: ≥95%) was procured from Aladdin Co., Ltd. (Shanghai, China); D-carvone (≥98.5%) was purchased from Shanghai Sangong Bioengineering Co., Ltd. (Shanghai, China); Tween 80, anhydrous ethanol, glacial acetic acid, and other reagents were purchased as analytical grade from Sinopharm Group Co., Ltd. (Beijing, China); *Staphylococcus aureus* (*S. aureus*, ATCC 6538) and *Escherichia coli* (*E. coli*, ATCC 8099) were obtained from Qingdao Hopebiol Co., Ltd. (Qingdao, China), and stored at −20 °C for subsequent use. Materials used for the ramen noodles (flour, salt, oil, etc.), as well as the plastic wrap (PE) and PP boxes, were purchased at the local market.

### 2.2. Preparation of Composite Film

The method of Wen et al. [21] was referenced with slight modifications to prepare composite membranes. Slowly add CS powder to an acetic acid aqueous solution (1%, *v*/*v*) to prepare a 2% (*w*/*v*) CS homogeneous solution and stir magnetically at 60 °C for 20 min to ensure thorough mixing. Similarly, prepare a 7% (*w*/*w*) homogeneous solution of gelatin using distilled water. Subsequently, 20 g of CS solution and 10 g of GEL solution were taken and mixed well, 1% Tween 80 and 2% glycerol by mass of the solution were added as lubricant and surfactant. D-carvone was subsequently added to the solution to give final concentrations of 0%, 0.5%, 1%, 2%, 3%, and 4% (based on “total solution mass”, *w*/*w*) in the composite membrane. The mixture was homogenized at 8000 g for 1 min. Subsequently, the solution was subjected to sonication to remove any air bubbles. This process was carried out at a temperature of 40 °C for a duration of 30 min to obtain the film-forming solution. Finally, pour the resulting film solution completely onto a polytetrafluoroethylene (PTFE) plate (20 × 20 cm) and dry it at 42 °C and 45 ± 5% relative humidity for 48 h to obtain the composite film. This procedure was undertaken to yield the composite films. The composite films were named as CS–GEL-0, CS–GEL-0.5D, CS–GEL-1D, CS–GEL-2D, CS–GEL-3D, and CS–GEL-4D, respectively.

### 2.3. Characterization of Composite Films

The composite films were characterized in terms of their full range of appearance and morphology, mechanical properties, antioxidant activity, and antibacterial activity. In addition, the characterization was carried out under room temperature conditions (25 ± 2 °C and 45 ± 5% relative humidity), unless otherwise specified.

#### 2.3.1. Appearance of Composite Film

The thickness of ten different locations at the center and periphery of the composite film was measured using a digital micrometer calibrated with 0–25 mm standard blocks (accuracy ± 0.001 mm, DL321025S, Deli Group Co., Ltd., Ningbo, China). Zero calibration was performed before measurement. The final result was taken as the average value.

The color of the composite films was determined by referring to the methodology outlined in the work of Roy et al. [22]. The film samples were cut to a size of 5 × 5 cm and placed under a colorimeter to determine the *L**, *a**, and *b**, and each film was measured in parallel six times. The total color difference (Δ*E*) was determined as follows (1):(1)$$\Delta E=\sqrt{{\left(L^{*}-L_{0}\right)}^{2}+{\left(a^{*}-a_{0}\right)}^{2}+{\left(b^{*}-b_{0}\right)}^{2}}$$
where $L_{0}$, $a_{0}$, and $b_{0}$ denote standard sample colorimetric reference values.

#### 2.3.2. Moisture Content, Solubility, and Swelling Rate (SR) of Composite Films

Moisture content and solubility were adopted from Kan et al. [7]. Strips of 10 × 10 mm films were weighed and dried at 105 °C until the weight was constant (approximately 6–8 h, weighed twice at 30 min intervals until the difference is ≤0.5 mg). The weight of the composite film before and after drying was measured to calculate the moisture content of the film. Immerse the dry film in distilled water at 25 °C overnight. After soaking, gently touch the surface of the film with filter paper to remove free water droplets, then air-dry at 25 °C for 5 min to avoid interference from residual moisture. The weight of the composite film before and after immersion was measured to calculate the solubility of the film. The formula is shown below:(2)$$Moisture\;content\;\left(\%\right)=\frac{W_{0}-W_{1}}{W_{0}}\times 100\%$$(3)$$Solubility\;\left(\%\right)=\frac{W_{1}-W_{2}}{W_{1}}\times 100\%$$
where $W_{0}$ denotes the initial mass of the membrane, mg; $W_{1}$ denotes the mass of the membrane after drying, mg; and $W_{2}$ denotes the mass of the membrane after immersion, mg.

The SR of the composite film was slightly modified according to the method of Yadav et al. [23]. The films were cut into squares of 20 × 20 mm, and their weight was measured. The films were then put into distilled water for 10 min, followed by drying in an oven at 105 °C, and the weight was again measured. *SR* was calculated using the formula in (6):(4)$$SR\;\left(\%\right)=\frac{M_{2}-M_{1}}{M_{1}}\times 100\%$$
where $M_{1}$ is the original weight of the film, mg, and $M_{2}$ is the weight of the expanded film, mg.

#### 2.3.3. Determination of Mechanical Properties of Composite Films

Mechanical properties were referenced from the original methods [24,25]. The films were cut to a size of 150 × 30 mm and fixed in an electronic universal testing machine (QJ210-50 N, Shanghai Qingji Instrumentation Technology Co., Shanghai, China) for the determination. The initial spacing was set to 50 mm, and the tensile speed was 1 mm/s. The tensile strength (TS) and elongation at break (EB) were calculated as follows:(5)$$TS=\frac{F}{S}$$
where *TS* indicates the tensile strength of the film, MPa; *F* indicates the film tensile force, N; and *S* indicates the cross-sectional area of the film (measured film width × sample width 30 mm), m^2^.(6)$$EB=\frac{L_{a}-L_{0}}{L_{0}}$$
where *EB* indicates the film elongation at break, %; *L_a_* indicates the elongation of the film at break, mm; and *L*_0_ indicates the initial length of the film, mm.

#### 2.3.4. Determination of Water Vapor Transmission Rate (WVTR) of Composite Films

According to the method of Liu et al. [26], the composite film was cut into circles with a diameter of approximately 10 cm. Before testing, all samples were equilibrated for 48 h in a standard environment of 25 °C and 45 ± 5% relative humidity to ensure that the moisture content was consistent with the test conditions. The WVTR was measured using a water vapor transmission rate tester (WVTR-I3, Guangzhou BoGu Experimental Equipment Co., Ltd., Guangzhou, China). The test conditions were as follows: temperature 38 °C, relative humidity 2%, and airflow 0.5 m/s for each sample, measured three times.

#### 2.3.5. Determination of Water Contact Angle (WCA) of Composite Films

The method in Chen et al. [27] was modified. A syringe was used to drop 5 μL of water onto the 2 × 2 cm membrane surface, and the WCA value was recorded after 30 s. And three different positions were taken to determine the hydrophobicity of the films.

#### 2.3.6. Scanning Electron Microscope (SEM)

Prepare and dry a 5 × 5 mm film. Then fix the paste with conductive glue and spray gold (use an ion sputter (20 mA, 90 s) to obtain a uniform 10 nm gold layer; all samples are processed in the same batch to ensure consistent film thickness). Observation was performed using a scanning electron microscope (S-3400N-II, Hitachi, Tokyo, Japan). Random observations were performed at 200 × and 5000 × magnification, respectively.

#### 2.3.7. Fourier Transform Infrared Spectroscopy (FTIR)

A Fourier Transform Infrared Spectrometer (FTIR) (Nicolet IS50, Thermo Fisher Scientific Inc., Waltham, MA, USA) was used to determine the changes in the chemical structure of the composite films. The samples were cut into 10 × 10 mm squares and measured in a wavelength range of 500–4000 cm^−1^ with a resolution of 4 cm^−1^ and scanned 32 times/s to characterize the differences in the changes in the surface groups of the samples.

#### 2.3.8. Thermal Properties

Cling film with better thermal stability can effectively maintain its structure and function in a wider temperature range and can significantly reduce the loss of properties caused by thermal degradation. Thermal properties of films were analyzed using a thermogravimetric analyzer (TG209F1, NETZSCH Instruments GmbH, Selb, Bavaria, Germany) according to the original method with slight modifications [28]. The TGA of the composite films was analyzed by chopping 5 mg of the composite films to a process of fragmentation in a crucible, followed by heating from 30 to 600 °C at a rate of 10 °C/min under N_2_ atmosphere (50 mL/min).

### 2.4. Antioxidant Properties of Composite Films

The antioxidant properties of the composite membrane were measured by DPPH and ABTS^+^ radical scavenging capacity. The 5 × 5 mm composite film was placed in ethanol and stored at room temperature away from light. After 60 min of reaction, 2 mL of the sample was collected (where Trolox at a concentration of 0.1 mg mL^−1^ was used as a positive control and pure ethanol as the solvent blank) and subjected to centrifugation at 6000× *g* for 1 min after slight shaking to obtain a clarified sample solution to be tested. The samples were analyzed according to the instructions of the kit (Shanghai Sangon Biotech Co., Ltd., Shanghai, China).

### 2.5. Determination of Antibacterial Activity of Composite Films

The antimicrobial properties of the composite films were assessed following Zhang et al. [29] with minor modifications. *E. coli* and *S. aureus* were activated at 37 °C in nutrient broth (NB) for 12 h and diluted to 10^6^ CFU mL^−1^. Take 1 mL of the bacterial suspension and mix it into 20 mL of 45 °C nutrient agar (NA) and pour into plates. A 6 mm diameter UV-sterilized (30 min) composite film disk was placed at the center of the plate. After incubation at 37 °C for 24 h, the diameter of the inhibition zone (mm) was measured.

### 2.6. Application of Composite Films in Xinjiang Ramen

#### 2.6.1. Preparation of Xinjiang Ramen

The dough was prepared with 1% saline in purified water, mixed with flour and saline in the ratio of 2:1, and then kneaded for 30 min to obtain the dough. The dough was wrapped with plastic wrap and allowed to rise at room temperature for 15 min, then kneaded for another 30 min, and finally kneaded into 20 cm long fresh noodles with a diameter of 2 cm and coated with a thin layer of edible oil to obtain Xinjiang ramen fresh noodles.

#### 2.6.2. Preservation of Freshness of Xinjiang Ramen by Composite Films

The above Xinjiang ramen fresh noodles were divided into 100 g portions and then wrapped with composite films (CS–GEL-0, CS–GEL-0.5D, CS–GEL-1D, CS–GEL-2D, CS–GEL-3D, CS–GEL-4D). The samples were put into a disposable plastic box, covered with a lid, and stored at room temperature (25 °C). The samples were taken at intervals of 0 d, 1 d, 2 d, 3 d, and 4 d for determination of the total number of colonies and sensory evaluation. Fresh Xinjiang ramen noodles wrapped with edible-grade PE plastic wrap were used as the control group, and fresh noodles wrapped without wrapping film were used as the blank group (the packaging method is shown in Figure 1).

#### 2.6.3. Determination of Total Colony Counts

The total colony count of fresh noodles of Xinjiang ramen was determined by the method of the Chinese national standard GB/T 4789.2-2022 [30]“National standard for food safety Microbiological examination of food Determination of total colony count”.

#### 2.6.4. Sensory Evaluation

The Chinese industry standard for the evaluation of ramen noodles was slightly modified based on LS/T 3202-1993 [31]“Wheat flour for noodles”. Fresh noodles packed in composite films (conducted after 2 days of storage; as shown in Section 2.6.3, the total colony count still complies with national standards at this point, and the noodles have not shown any obvious deterioration) were stretched to three times their original length, cooked in boiling water for 5 min, quickly removed, and placed in water below 22 °C for 10 s to cool down before placing them on a white porcelain plate. A sensory evaluation panel consisting of 7 trained food professional evaluators (4 females and 3 males, aged 25–40 years old, who underwent training one week prior to the experiment) rated the wet noodles on a sensory composite scale of color (10 points), morphology (10 points), palatability (20 points), toughness (20 points), viscosity (20 points), smoothness (5 points), and taste (15 points). The evaluation was conducted using a blind tasting method in a standard sensory evaluation laboratory.

### 2.7. Statistical Analysis

All experimental data were expressed as mean ± standard deviation (n = 3). Duncan’s test was performed using SPSS-27 software, and differences between groups were compared by one-way ANOVA (α = 0.05), with *p* < 0.05 considered statistically significant. The software used for visualization was Origin 2021 (OriginLab, Northampton, MA, USA).

## 3. Results and Discussion

### 3.1. Appearance Characteristics of Composite Films

A robust correlation has been demonstrated between consumer acceptance and the overall appearance of food packaging [32,33]. As illustrated in Figure 2A, the appearance of the composite films is demonstrated. Firstly, the thickness of the composite film, which could not be separated by the naked eye, is summarized in Figure 2B. In comparison with the control group, the thickness of the composite films increased from 0.141 mm to 0.302 mm as the D-carvone content increased. The increase in the thickness of the composite films was found to be non-significant (*p* < 0.05) at an additive amount of 0.5%. However, when the additive amount exceeded 1%, a significant increase in thickness was observed (*p* < 0.05), concomitant with an increase in additive amount. The thickness of the composite film was related to its composition, process, and drying temperature, and the increase in the amount of D-carvone added may have changed the structure of the composite films, resulting in an increase in the thickness, which can be confirmed by the subsequent results of the SEM images and FTIR, etc. In addition, the increase in D-carvone increased the spatial distance within the membrane matrix, thus making the film thicker [8]. Haghighi et al. [12] obtained similar results for CS–GEL composite films, where the composite film thickness increased with the addition of essential oil content of the plant. Increasing thickness has a double-edged effect. Increasing thickness enhances puncture resistance and makes the film more suitable for transportation and stacking, but excessive thickness (>0.25 mm) can cause the composite film to bend 180° and crack, and WVTR will also increase due to pores (see Section 3.4), thereby affecting the performance of the film.

In order to analyze the color of the composite film, firstly, the appearance of the film (Figure 2A) shows that the CS–GEL-0 film is almost transparent, while CS–GEL-D is light yellow. The color becomes darker and more yellow with the increase in D-carvone content, which is because D-carvone itself has a distinct yellow color. In Table 1, the color change of the composite film was reflected by the change in *L**, *a**, and *b** values, and Δ*E*. The addition of D-carvone resulted in a significant decrease (*p* < 0.05) in the *L** value of the composite films as compared to the control, and with the increase in the amount of the addition, the composite films also showed a significant increase in the *a** and *b** values (*p* < 0.05), with an overall increase in the corresponding Δ*E* values. Other researchers have reported similar results where the addition of ginger essential oil, which is yellowish, resulted in lower *L** and higher *b** values for CS–GEL films [34]. An assistant professor mentioned that 75% of consumers pay attention to label colors, and warm colors (red/orange) are associated with sweetness, while cool colors (blue/green) are associated with sourness and health [35]. It can be seen that the light yellow color of the CS–GEL-D film helps to increase the willingness to purchase.

### 3.2. Moisture Content, Solubility, and Swelling Analysis of Composite Films

The moisture content of packaging films is one of the key factors affecting the performance of packaged products. CS-GLE composite films may absorb water and expand in high-humidity environments, reducing their strength and affecting their protective function [36,37], and the moisture content can reflect the total amount of free volume space filled by water molecules in the film network system [12]. As shown in Figure 2C, when D-carvone ≤1%, the moisture content of the composite membrane slightly increases, which can be attributed to the large number of hydrophilic groups in CS and GEL [38]; when it is >1%, D-carvone forms additional hydrogen bonds and hydrophobic associations with CS–GEL, occupying free volume and reducing accessible sites for water molecules, thereby decreasing the water content [39,40]. Kan et al. [7] reported that the addition of a plant extract to CS–GEL hybrid membranes (*Crataegus pinnatifida* fruit extract) resulted in a slight decrease in water content.

However, a decrease in moisture content does not necessarily lead to a decrease in solubility. Figure 2D shows that solubility slightly increases with the addition of D-carvone, which may be attributed to the hydrophobic terpenoid skeleton of D-carvone inserting into the CS–GEL network, thereby weakening the density of intermolecular hydrogen bonds and facilitating water molecule penetration to disrupt interchain interactions [41]. Additionally, hydrophobic microdomains form “defects” at the interface, increasing the probability of releasing soluble oligomers or monomers, resulting in a slight increase in solubility [42].

Meanwhile, the reduced network crosslinking density also makes the membrane structure more loosely packed, leading to an increase in swelling rate from 41.85% to 49.05%, as shown in Figure 2E. In summary, D-carvone exhibits both “hydrophobic displacement” and “weakened crosslinking” effects: the former reduces water absorption (lowering water content), while the latter increases water permeability and chain segment mobility (enhancing solubility and swelling rate). When the swelling rate is around 45%, it can moderately prolong the release time of active substances; if further increased, it may be detrimental to practical applications due to reduced membrane mechanical strength [43].

### 3.3. Mechanical Properties of Composite Films

Mechanical properties are one of the most basic and important properties of packaging films, and good mechanical properties can ensure that they remain intact during transportation as well as withstand external forces better. The mechanical properties of composite films include TS and EB, as shown in Figure 3A,B. With the increase in D-carvone content, the TS of the composite films showed an increasing and then decreasing trend. When the addition amount was less than 1%, the TS increased from 11.38 Mpa to 15.71 Mpa; when it was greater than 2%, the TS of the composite films decreased significantly (*p* < 0.05) from 12.38 Mpa to 3.9 Mpa. Both CS and GEL belong to biopolymers, and the two are capable of combining to form a network structure through hydrogen bonding [44]. As a small molecule, adding too much D-carvone will lead to poor dispersion, thus destroying the ordered structure of the polymer molecular chain and hindering the formation of a crosslinked network. In addition, the ring-external ketone group of D-carvone competes with the -NH_2_ of CS for hydrogen bonds. When present in excess, it forms a “hydrophobic cluster” that disrupts the three-dimensional network of CS–GEL, leading to phase separation and the formation of pores. Therefore, the more D-carvone added, the less the film’s ability to resist tensile stress, leading to a decrease in its overall TS [45]. This trend is similar to that observed by Qian et al. [2], who doped anthocyanin and essential oil of carvone in CS films. EB shows the opposite result, showing a decreasing and then increasing trend, which may be caused by the addition of D-carvone, which affects the plasticization of water in the composite material, mirroring the same trend as the water content in the composite film.

### 3.4. WVTR and WCA Analysis of Composite Films

The barrier property of the film is reflected by determining the WVTR of the composite film. The barrier properties of packaging materials affect the shelf life of products. A lower WVTR value indicates that the film material has stronger waterproof properties, while a high WVTR may lead to condensation accumulation inside the packaging, increasing water activity and accelerating the growth of mold and bacteria [46]. The results in Figure 3C indicate that when the addition of D-carvone is ≤1%, the WVTR of the composite film decreases; however, as the amount of D-carvone continues to increase, the WVTR paradoxically increases. Within the formulation range of 0.5% to 3%, the WVTR is significantly lower than that of the control group (*p* < 0.05). The addition of a small amount of D-carvone can introduce hydrophobic groups, which can improve the hydrophobicity of the composite film and reduce the water vapor transmission rate. However, excessive D-carvone is unevenly dispersed in the film, easily forming micro-pores or gaps, making the composite film structure loose and the bond between the components no longer dense, ultimately leading to a decrease in the film’s barrier properties against water molecules [47].

Figure 3D shows the WAC of the composite films, an increase in WCA can reduce bacterial adhesion, but if WCA > 80°, condensation is likely to form beads and slide onto the food surface, thereby increasing the risk of cross-contamination. Addition of D-carvone increases the WAC of the composite films relative to that of CS–GEL-0, indicating that the surface of the composite film has a higher hydrophilicity. The WAC of CS–GEL-2D was 72 (θ > 65), indicating that the composite film exhibited a high WAC value and a robust hydrophobic surface. Studies have demonstrated that the WAC of CS films ranged from 88 to 100 [48], which was predominantly attributed to the formation of robust hydrogen bonds between the CS molecular chains, thereby impeding the penetration of water molecules. The addition of GEL resulted in a significant decrease in the WCA value of CS–GEL. This phenomenon can be attributed to the presence of hydrophilic groups within GEL, which enhance the interaction of CS–GEL with water, thereby reducing its surface hydrophobicity [49]. The WCA decreased when the D-carvone content was higher than 3%, which may be attributed to the fact that D-carvone itself is hydrophobic, and higher concentrations may lead to intermolecular agglomeration.

### 3.5. Microstructure of Composite Film

SEM can clearly reflect the microstructure of the composite film, and its internal microstructure can reflect the compatibility and miscibility between the molecules in the film matrix, which can directly affect the performance of the material [12]. As shown in Figure 4A(a1–f1), the surface of the composite film was dense and homogeneous in CS–GEL-0, CS–GEL-0.5D, and CS–GEL-1D, and no obvious pores were observed. With the increase in D-carvone, the surface of the films showed uneven folds, which may be due to the increase in the addition amount, which caused the agglomeration phenomenon and reduced the interaction with CS–GEL, resulting in the occurrence of the “swelling” phenomenon. SEM 5000× Figure 4A(a2–f2) shows that when ≤1% D-carvone is present, the film surface is dense and pore-free, with a tensile strength of 15.71 MPa and WVTR of 174.18 g m^−2^ d^−1^. A small amount of D-carvone is tightly bound to the base film material through hydrogen bonding, making the composite film surface uniform, without pores or wrinkles. At >2%, micro-pores and rough wrinkles appear, corresponding to a decrease in tensile strength to 3.90 MPa and an increase in WVTR to 265.65 g m^−2^ d^−1^. The micro-pores provide diffusion pathways for water vapor and microorganisms, resulting in a simultaneous decline in barrier performance and mechanical properties. In addition, it can be inferred that the molecular insertion weakened the interchain hydrogen bonds of CS–GEL, thereby reducing the local order. This phenomenon is consistent with the “internal plasticization of essential oil small molecules” mechanism reported by Qian et al. (2025) [2].

### 3.6. FTIR Analysis

The FT-IR spectra of the composite films are shown in Figure 4B. The CS–GEL film has significant characteristic peaks at 3400 cm^−1^, 1650 cm^−1^, and 1550 cm^−1^, which are related to the stretching vibration of the O–H bond [25]. The characteristic peaks of the D-carvone were mainly the stretching vibration of C=O near 1700 cm^−1^. The peak shapes and intensities of the characteristic peaks at 3400 cm^−1^, 1650 cm^−1,^ and 1550 cm^−1^ were gradually enhanced with the increase in D-carvone addition. Among these, the introduction of D-carvone resulted in broadening of the peak shape at 3400 cm^−1^ (O-H/N-H) and a shift toward lower wavenumbers, while the intensity of the peak at 1728 cm^−1^ to 1572 cm^−1^ (C=O amide I) and 1572 cm^−1^ to 1471 cm^−1^ (amide II) decreases, indicating that the ketone group forms additional hydrogen bonds with the -NH_2_ group of CS and the -CO-NH- group of GEL. It can be confirmed that D-carvone is effectively loaded in the CS–GEL base film.

### 3.7. Thermal Properties

The thermogravimetric (TGA) curves and derivative thermogravimetric (DTG) curves of the composite films are shown in Figure 4C,D. In the temperature range of 100–200 °C, the samples showed a small amount of mass loss, which accounted for about 2.0–3.5% of the total mass, probably due to the evaporation of water and small molecules in the composite film as a result of the heating process. The temperature range of 200–400 °C is the main decomposition stage of the composite film. Among them, the decomposition of CS–GEL was mainly due to sugar chain breakage and protein degradation.

From the DTG curves, it can be seen that the main peak temperature of CS–GEL-0 was 380 °C, and after the addition of D-carvone, the main peak temperature decreased to 350 °C at the lowest level, which might be since the addition of D-carvone affected the hydrogen bonding between the CS and reduced the cross-linking between the molecules. Compared with CS–GEL-0, the addition of D-carvone accelerated the decomposition of the base film material, resulting in an increase in the loss rate of the composite films from 24% to 26–34%. At the high temperature stage above 400 °C, the thermal decomposition temperature of the CS–GEL-4D film decreased by about 4 °C, probably because D-carvone interfered with the intermolecular interactions of the CS–GEL matrix related to the phase separation of the base film material. The residual mass decreased from 15.65% to 14.37% at this stage of addition of less than 1%, but the residual mass of CS–GEL-4D increased to 16.39%, indicating that the addition of low concentrations of D-carvone did not appear to significantly disrupt the structure of the polymer network but made the intermolecular interactions enhanced and was able to improve the localized thermal stability of composite films. Therefore, D-carvone containing less than 1% of D-carvone can form hydrogen bonds with CS and GEL and enhance the interactions between the components, with better thermal stability.

### 3.8. Antioxidant Activity of Composite Films

The antioxidant activities of the composite membranes were as shown in Figure 5A,B. The ABTS^+^ and DPPH values of CS–GEL membranes were 12.78% and 11.78%, respectively, which may be attributed to the antioxidant activity of CS. Compared with CS–GEL-0, the ABTS^+^ and DPPH of the composite films increased to 21.4–46.57% and 17.23–41.95%, respectively, with the increase in D-carvone addition; and the degree of increase in ABTS^+^ was more pronounced. ABTS^+^ relies mainly on electron transfer capacity, whereas DPPH emphasizes more emphasis on hydrogen-donating capacity. The chemical structure of D-carvone may be more adapted to the electron transfer pathway of ABTS^+^, leading to a higher scavenging efficiency, a result that is consistent with the pattern of phenolic compounds reported in the literature to exhibit differences in performance in different antioxidant systems [50]. In addition, compared with the results of existing literature, its antioxidant performance is also relatively superior, making it very suitable for use in environments with high oil and moisture content [14,15]. The antioxidant capacity of CS–GEL-3D and CS–GEL-4D films increased more slowly than that of CS–GEL-0 to CS–GEL-2D, which may be related to the limited solubility of D-carvone. Oversaturation and precipitation reduced the effective concentration, leading to a slowdown in the growth of free radical scavenging rate.

### 3.9. Antibacterial Activity

The antibacterial properties were explored by comparing the inhibition zone size of the composite membranes. From Table 2 and Figure 5C, the inhibition zone of CS–GEL for *E. coli* and *S. aureus* was 8.67 ± 0.54 mm and 8.73 ± 0.34 mm, respectively. CS had a certain degree of bacterial inhibitory property, with no bacterial growth on the contact surface of the composite films, and a clear boundary of the inhibition zone, which was in agreement with the report [51]. The inhibition zone of *E. coli* in CS–GEL-0.5D was 8.96 ± 0.13 mm, which did not differ significantly from CS–GEL-0 (*p* > 0.05), while that of *S. aureus* was 9.69 ± 0.60 mm, which was slightly larger than that of *E. coli*. The inhibition zone of *E. coli* increased from 11.54 ± 0.55 mm to 24.56 ± 1.00 mm with the addition of D-carvone, whereas that of *S. aureus* increased from 10.11 ± 0.51 mm to 27.83 ± 1.34 mm. The differences in the cell wall structures between them were related to the fact that D-carvone molecules were more readily adsorbed into the multilayered peptidoglycan structure of *S. aureus* peptidoglycan structure, whereas the outer membrane lipopolysaccharide layer of Gram-negative bacteria may hinder the penetration of active ingredients [18]. In addition, the lipophilic nature of carvone may make it easier to penetrate the cell membrane of *S. aureus*, thus enhancing the bactericidal efficiency [16]. This result is consistent with the mechanism of selective inhibition of Gram-positive bacteria by terpenoids reported in the literature [52]. In summary, D-carvone loading on the composite membrane was able to improve the antimicrobial properties of the composite film.

### 3.10. Application of Composite Film in Xinjiang Ramen

From the above-mentioned properties of the composite film, it can be understood that it has excellent performance. Therefore, the further freshness preservation efficacy of the composite film in the real food system was verified by simulating the storage environment of Xinjiang ramen with high moisture and lipid (shown in Figure 6).

#### 3.10.1. Changes in the Total Number of Colonies During Storage

Changes in the total number of colonies in Xinjiang ramen at a storage temperature of 25 °C with the composite film further illustrated the freshness preservation effect of the composite film (Table 3).

At 1 d of storage, only a small amount of microbial growth was observed in Xinjiang ramen wrapped in PE packaging film, and at 2 d, no colony growth was observed in Xinjiang ramen wrapped in CS–GEL-1D, CS–GEL-2D, and CS–GEL-3D, which indicated that the microbial growth and reproduction were effectively inhibited. With the increase in storage days, the total number of colonies of fresh noodles in each treatment group increased significantly, but the total number of colonies of CS–GEL-1D, CS–GEL-2D, and CS–GEL-3D treatments still complied with the acceptable microbial index limit values in the Chinese national standard GB19295-2021 [53]“National Standard for Food Safety Frozen Noodles, Rice and Prepared Foods”. When stored for 4 d, the PP-packed blank group and the fresh noodles of CS–GEL-0 were already moldy. Surprisingly, there was a higher colony count after 4 d of CS–GEL-4D treatment. This may be related to the performance of the composite film. Consistent with the above film performance results, the high D-carvone content caused the WVTR to rebound to 265.65 g m^−2^ d^−1^, reducing the barrier properties of the composite film and increasing the surface water activity, which easily led to microbial contamination, loss of sustained antibacterial activity, and “rebound” contamination. Therefore, in this study, 1% and 2% D-carvone composite films can control the total colony count to <5 log CFU g^−1^ and extend the shelf life to about 4 days.

#### 3.10.2. Effects on the Sensory Quality of Xinjiang Ramen

Fresh Xinjiang ramen stored for two days (the total number of colonies complied with the Chinese national standard GB19295-2021) was subjected to steaming treatment, and the sensory qualities of the noodles were compared over different storage periods. As shown in Figure 6A, from the overall appearance, the color of Xinjiang ramen wrapped in CS–GEL-4D film was slightly dark and grayish, which was not much different from other packages. In addition, the surface of the fresh noodles was slightly gray after cooking, which might be due to the poor barrier property of the composite film, leading to oxidation of the fresh noodles. Figure 6C reflects the sensory scores of Xinjiang ramen in different packages after cooking, and there was little difference between the Xinjiang ramen packaged with CS–GEL-0, CS–GEL-0.5D, CS–GEL-1D, and PE plastic film in terms of texture, with the sensory scores being higher. Xinjiang ramen wrapped in these films had a wheat flavor, smoothness, strength, and elasticity. Xinjiang ramen wrapped with no film in a PP plastic box had a slightly hard texture, and the rest of the performance was moderate. Xinjiang ramen wrapped with CS–GEL-2D, CS–GEL-3D, and CS–GEL-4D had the characteristic odor of D-carvone, and the higher the content, the more obvious the odor was, resulting in lower organoleptic scores. Therefore, CS–GEL-1D and CS–GEL-2D had a better preservation effect on fresh noodles, exhibited a certain antibacterial effect, and had little effect on the taste of the product itself.

## 4. Conclusions

This study successfully prepared a CS–GEL-D composite film, which exhibits both antioxidant and antibacterial functions. In terms of performance, as the content of D-carvone increases, the thickness and water contact angle of the composite film increase, while the tensile strength first increases and then decreases. D-carvone can enhance the hydrogen-bonding effect of the composite film. When the addition amount is ≤2%, the composite film is evenly dispersed, has good thermal stability, and has low water vapor permeability. When the addition level exceeds 2%, performance declines—thermal stability decreases; antioxidant benefits slow down; antibacterial effects decrease; and the volatile odor reduces consumer ratings. In terms of antioxidant and antibacterial activity, the composite film can effectively improve the antioxidant and antibacterial activity against *E. coli* and *S. aureus*. In Xinjiang ramen preservation application, the composite film with 1–3% D-carvone has an antibacterial effect, but contents above 3% negatively affect flavor, resulting in a lower sensory score. Considering the freshness preservation effect, 1–2% D-carvone was found to be appropriate. This study provides strong support for the application of natural terpenoid-based composite films in the field of natural food preservation, which is helpful for the development of efficient and safe food additives and packaging materials.

## Figures and Tables

**Figure 1 foods-14-02645-f001:**
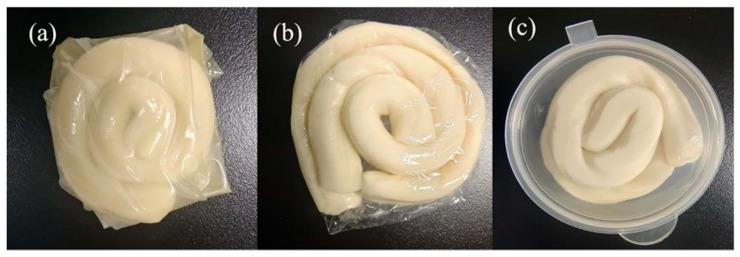
Packaging for Xinjiang ramen. (**a**) Composite film packaging Xinjiang ramen; (**b**) PE cling film packaging Xinjiang ramen; (**c**) PP cling box packaging Xinjiang ramen.

**Figure 2 foods-14-02645-f002:**
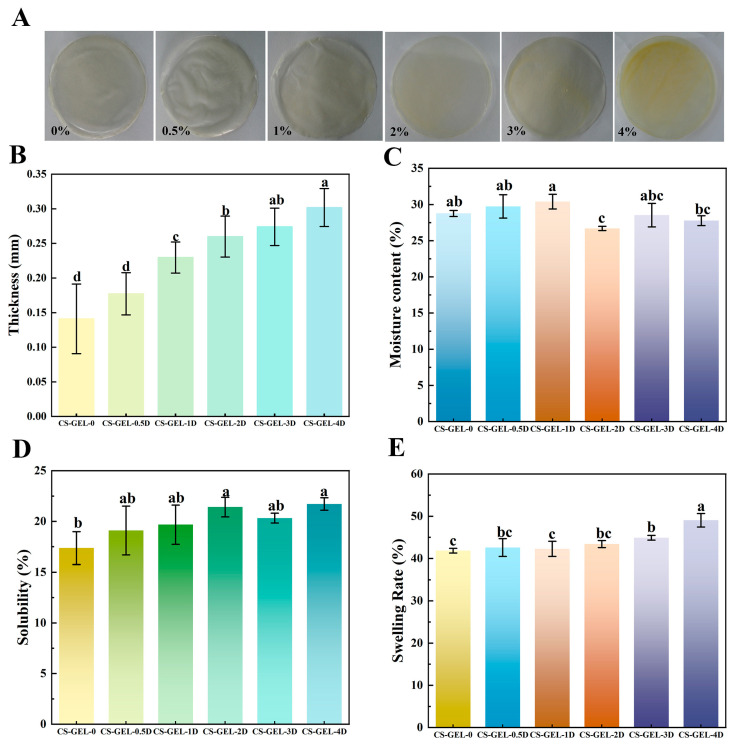
(**A**) Appearance of the composite film (0–4% expression of D-carvone addition); (**B**–**E**) indicate the thickness, moisture content, solubility, and swelling rate of the composite film, respectively. Where CS–GEL-0 to CS–GEL-4D denote composite films to which 0 to 4% content of D-carvone has been added, respectively. Note: Different lowercase letters in the table indicate significant differences (*p* < 0.05), where a–d indicates that the differences range from high to low.

**Figure 3 foods-14-02645-f003:**
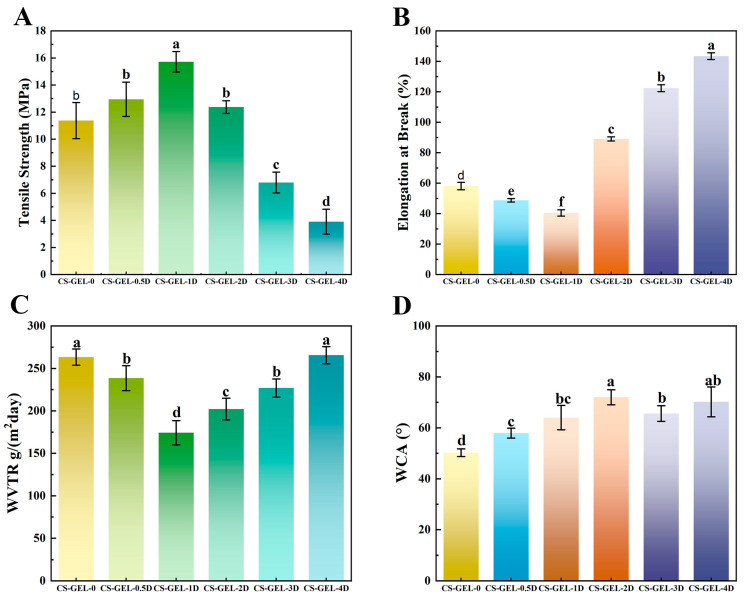
Tensile strength (**A**), elongation at break (**B**), water vapor transmission rate (**C**), and water contact angle (**D**) of composite films. Where CS–GEL-0 to CS–GEL-4D denote composite films to which 0 to 4% content of D-carvone has been added, respectively. Note: Different lowercase letters in the table indicate significant differences (*p* < 0.05), where a–d indicate that the differences taper from high to low.

**Figure 4 foods-14-02645-f004:**
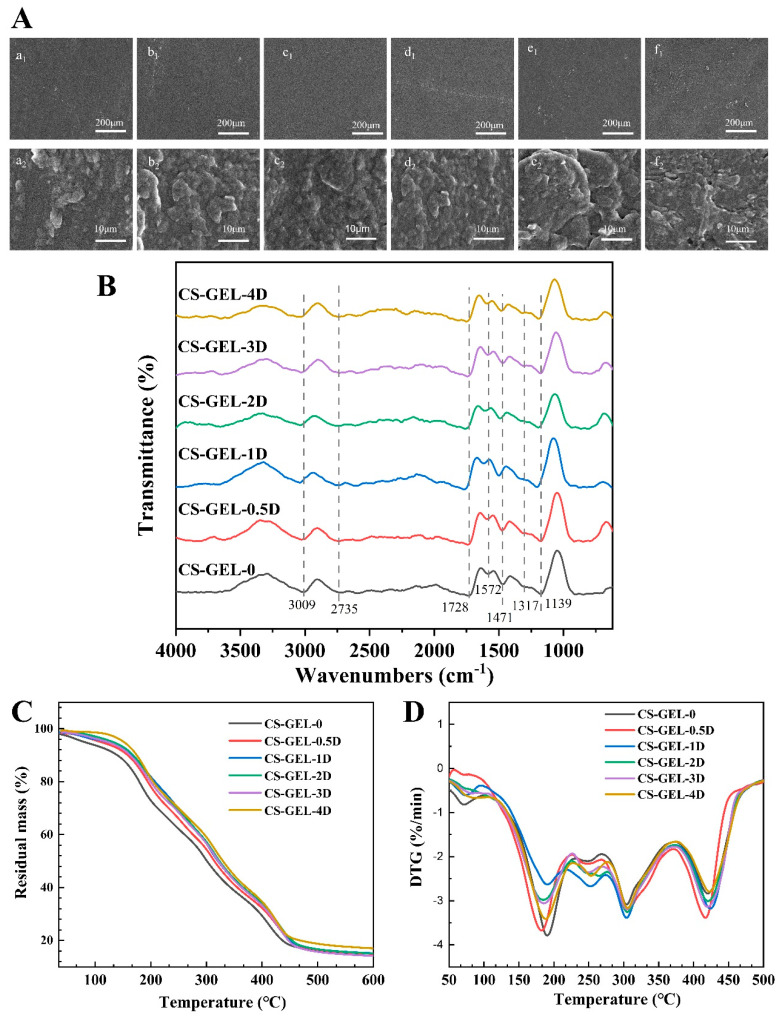
SEM of the composite films (**A**), where “a–f” represent CS–GEL composite films with 0%, 0.5%, 1%, 2%, 3%, and 4% D-carvone content, respectively, and “1 and 2” represent magnification of 200× and 5000×, respectively. FTIR spectra of the composite films (**B**). Thermogravimetric analysis (**C**) and differential thermogravimetric curve (**D**) of the composite films. Where CS–GEL-0 to CS–GEL-4D denote composite films to which 0 to 4% content of D-carvone has been added, respectively.

**Figure 5 foods-14-02645-f005:**
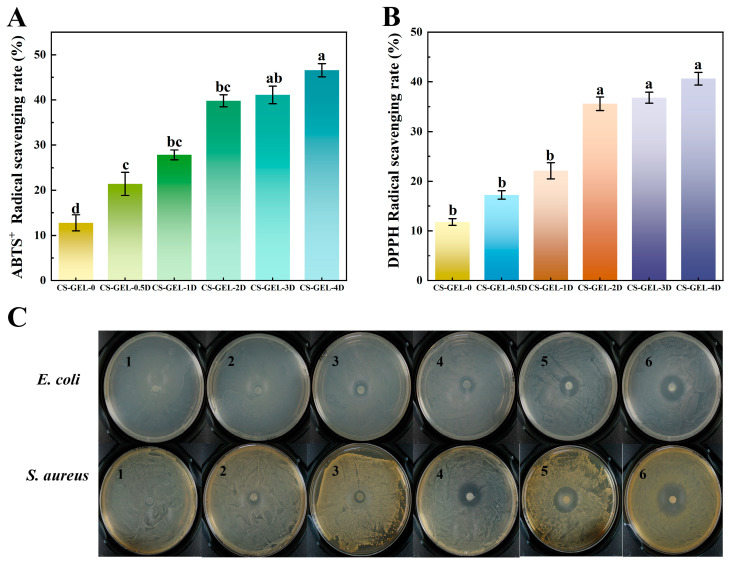
ABTS^+^ radical scavenging (**A**) and DPPH radical scavenging (**B**) of composite films. Experiments on the inhibition zone of composite films (**C**). Where CS–GEL-0 to CS–GEL-4D denote composite films to which 0 to 4% content of D-carvone has been added, respectively (1: CS–GEL-0; 2: CS–GEL-0.5D; 3: CS–GEL-1D; 4: CS–GEL-2D; 5: CS–GEL-3D; 6: CS–GEL-4D). Note: Different lowercase letters in the table indicate significant differences (*p* < 0.05), where a–d indicate that the differences taper from high to low.

**Figure 6 foods-14-02645-f006:**
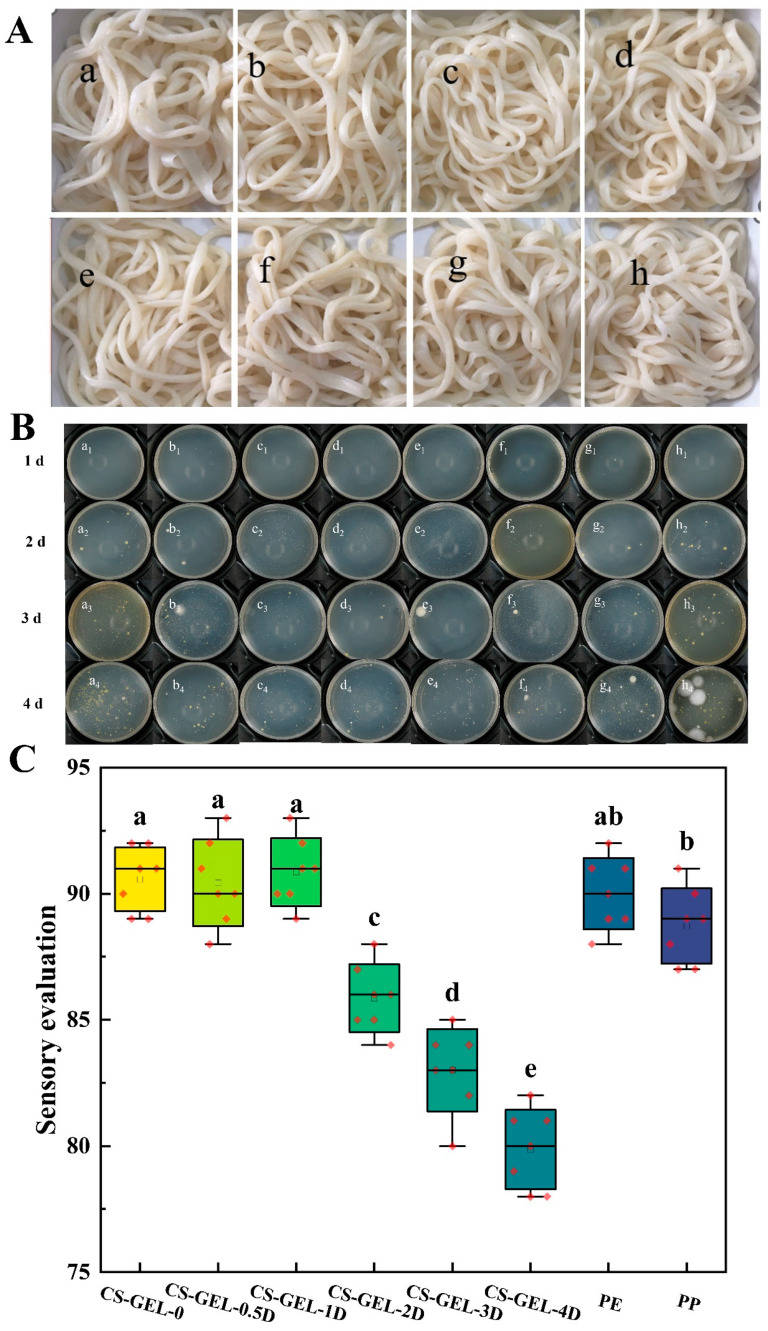
Effect of different packages on Xinjiang ramen during storage (**A**), where “a–f” represent CS–GEL composite films with 0%, 0.5%, 1%, 2%, 3%, and 4% D-carvone content, respectively, and “g and h” represent PE cling film packaging and PP cling box packaging, respectively. Changes in the total number of colonies of Xinjiang ramen packed in composite films during storage (**B**). Sensory scores of Xinjiang ramen in different packages (**C**). Where CS–GEL-0 to CS–GEL-4D denote composite films to which 0 to 4% content of D-carvone has been added, respectively. Note: Different lowercase letters in the table indicate significant differences (*p* < 0.05), where a–e indicate that the differences taper from high to low.

**Table 1 foods-14-02645-t001:** Color of laminated films.

Sample	Color Swatch	*L**	*a**	*b**	Δ*E*
CS–GEL-0		95.16 ± 0.94 ^a^	−3.23 ± 1.09 ^d^	13.76 ± 0.64 ^c^	14.23 ± 0.85 ^c^
CS–GEL-0.5D		93.29 ± 0.39 ^b^	−2.49 ± 0.14 ^c^	12.64 ± 1.22 ^c^	13.61 ± 1.19 ^c^
CS–GEL-1D		92.03 ± 0.51 ^c^	−2.33 ± 0.12 ^bc^	17.85 ± 0.36 ^b^	18.85 ± 0.50 ^b^
CS–GEL-2D		89.35 ± 0.62 ^e^	−1.87 ± 0.20 ^ab^	22.10 ± 2.85 ^a^	23.77 ± 2.63 ^a^
CS–GEL-3D		89.79 ± 0.36 ^de^	−1.40 ± 0.09 ^a^	21.07 ± 0.87 ^a^	22.59 ± 0.93 ^a^
CS–GEL-4D		90.17 ± 0.30 ^d^	−2.20 ± 0.06 ^bc^	21.77 ± 0.60 ^a^	23.14 ± 0.65 ^a^

Where CS–GEL-0 to CS–GEL-4D denote composite films to which 0 to 4% content of D-carvone has been added, respectively. Note: Different lowercase letters in the table indicate significant differences (*p* < 0.05), where a–d indicate that the differences taper from high to low. Color swatch: obtained from https://www.colortell.com/findlab (accessed on 6 May 2025) and based on *L** values, *a** values, and *b** values.

**Table 2 foods-14-02645-t002:** Experiments on the inhibition zone of composite films (mm).

Sample	*E. coli*	*S. aureus*
CS–GEL-0	8.67 ± 0.54 ^e^	8.73 ± 0.34 ^e^
CS–GEL-0.5D	8.96 ± 0.13 ^e^	9.69 ± 0.60 ^de^
CS–GEL-1D	11.54 ± 0.55 ^d^	10.11 ± 0.51 ^d^
CS–GEL-2D	13.43 ± 0.97 ^c^	16.60 ± 0.88 ^c^
CS–GEL-3D	20.07 ± 1.07 ^b^	22.87 ± 0.45 ^b^
CS–GEL-4D	24.56 ± 1.00 ^a^	27.83 ± 1.34 ^a^

Where CS–GEL-0 to CS–GEL-4D denote composite films to which 0 to 4% content of D-carvone has been added, respectively. Note: Different lowercase letters in the table indicate significant differences (*p* < 0.05), where a–e indicate that the differences taper from high to low.

**Table 3 foods-14-02645-t003:** Total colony count of fresh Xinjiang ramen noodles in different packages (log CFU g^−1^).

Sample	Storage Days			
	1 Day	2 Day	3 Day	4 Day
CS–GEL-0	0 ^b^	1.67 ± 2.08 ^cd^	14.33 ± 3.21 ^b^	41.67 ± 4.04 ^a^
CS–GEL-0.5D	0 ^b^	2.00 ± 1.00 ^bc^	6.00 ± 2.65 ^c^	13.33 ± 3.51 ^c^
CS–GEL-1D	0 ^b^	0 ^d^	1.00 ± 1.00 ^d^	4.00 ± 2.00 ^d^
CS–GEL-2D	0 ^b^	0 ^d^	2.67 ± 1.53 ^cd^	4.67 ± 1.53 ^d^
CS–GEL-3D	0 ^b^	0 ^d^	1.67 ± 2.08 ^d^	2.67 ± 1.53 ^d^
CS–GEL-4D	0 ^b^	0.67 ± 0.58 ^cd^	2.67 ± 1.53 ^cd^	11.00 ± 2.65 ^c^
PE	0.33 ± 0.42 ^a^	3.67 ± 1.53 ^b^	6.00 ± 3.00 ^c^	24.33 ± 5.03 ^b^
PP	0 ^b^	5.67 ± 1.53 ^a^	19.00 ± 2.65 ^a^	-

Where CS–GEL-0 to CS–GEL-4D denote composite films to which 0 to 4% content of D-carvone has been added, respectively. Note: Different lowercase letters in the table indicate significant differences (*p* < 0.05), where a–d indicate that the differences taper from high to low; 0 means not detected; – means fully molded exceeding the counting threshold.

## Data Availability

The original contributions presented in this study are included in the article. Further inquiries can be directed to the corresponding author.

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
