# Peer review of "Preparation and Characterization of D-Carvone-Doped Chitosan–Gelatin Bifunctional (Antioxidant and Antibacterial Properties) Film and Its Application in Xinjiang Ramen"

_foods, 2025, doi:10.3390/foods14152645_

Round 1
Reviewer 1 Report
Comments and Suggestions for Authors
ID: foods-3762765
Title: Preparation, characterization and application of D-carvone-doped chitosan-gelatin bifunctional films for antioxidant and antibacterial
Dear Editor
The manuscript is nice and the idea of incorporation D-carvone into chitosan-gelatin film is well. However, there are some points which must be solved.
- The title and keywords should reflect the experiments. This film was used for quality improvement of noodle.
- Abstract: please insert more details about polymer concentration, ratio, D-carvone concentrations and some quantitative data.
- Abstract line 19: Did you use this active film for improving quality of noodle? Please insert more information in abstract and Title
- Introduction: the objective of the study was not clearly stated. Why Xinjiang ramen? Please explained about the ramen problems and why edible films are useful for its preservation.
- Determination of mechanical properties, Line 179, TS and EB: please insert the abbreviation after first appearance.
- Antimicrobial activities: insert more information about cultures.
- Moisture content and solubility: why the results of moisture content and solubility had different trend? Why solubility increased by increasing D-carvone? Also, swelling increased by enhancing D-carvone content. Please explained in depth.
- WVTR, line 362: “WVTR increased by increasing D-carvone”. Please discuss and explain in depth.
- Based on which reference you choose D-carvone concentration?
- Please insert Figure 6.
- Please define treatments in Tables and Figure.
Author Response
Dear Editor and reviewers,
We appreciate you for taking the time to review our manuscript entitled "Preparation, characterization and application of D-carvone-doped chitosan-gelatin bifunctional films for antioxidant and antibacterial" (Manuscript ID: foods-3762765). We have carefully revised the manuscript, and the issues raised by the editor and reviewers have been addressed as follows. All the changes are marked in red fonts throughout the revised manuscript. Page/Line numbers, Figure and Table also refer to the revised manuscript.
With Best Regards,
Hong Wu, Ph.D.
Professor
School of Food Science and Engineering
South China University of Technology
Guangzhou, China
Email: bbhwu@scut.edu.cn (H. Wu)
Responds to the editor and reviewers’ comments:
Reviewer #1: The manuscript is nice and the idea of incorporation D-carvone into chitosan-gelatin film is well. However, there are some points which must be solved.
Comment 1: The title and keywords should reflect the experiments. This film was used for quality improvement of noodle.
Revision: We have revised the title and keywords based on your comments (Line 3-4, 27).
Comment 2: Abstract: please insert more details about polymer concentration, ratio, D-carvone concentrations and some quantitative data.
Revision: We have revised the Abstract based on your comments (Line 8-11).
Comment 3: Did you use this active film for improving quality of noodle? Please insert more information in abstract and Title
Revision: We have revised the abstract and Title based on your comments (Line 2-4, 17-19).
Comment 4: Introduction: the objective of the study was not clearly stated. Why Xinjiang ramen? Please explained about the ramen problems and why edible films are useful for its preservation.
Revision: We have revised the MS based on your comments (Line 102-108, 109-119).
Comment 5: Determination of mechanical properties, Line 179, TS and EB: please insert the abbreviation after first appearance.
Revision: We have revised the MS based on your comments (Line 188-189).
Comment 6: Antimicrobial activities: insert more information about cultures.
Revision: We have revised the MS based on your comments (Line 242-247).
Comment 7: Moisture content and solubility: why the results of moisture content and solubility had different trend? Why solubility increased by increasing D-carvone? Also, swelling increased by enhancing D-carvone content. Please explained in depth.
Revision: We have revised the MS based on your comments (Line 342-370).
Comment 8: WVTR, line 362: “WVTR increased by increasing D-carvone”. Please discuss and explain in depth.
Revision: We have revised the MS based on your comments (Line 401-414).
Comment 9: Based on which reference you choose D-carvone concentration?
Revision: Based on existing literature and preliminary experiments, we found that adding a small amount of additive significantly improves the hydrophilicity of the matrix, but adding less than 5% causes a rapid decline in the performance of the composite membrane, which requires nano-encapsulation technology to solve. We have also added explanations in the appropriate places in the MS (Line 98-102).
Comment 10: Please insert Figure 6.
Revision: We have revised the MS based on your comments (Line 548-556).
Comment 11: Please define treatments in Tables and Figure.
Revision: We have revised the MS based on your comments (Line 314-318, 335-340, 393-398, 451-456, 513-517, 539-542, 548-556, 562-564).
Reviewer 2 Report
Comments and Suggestions for Authors
comments attached in a pdf file

English language must be revised and improved. Specific examples of such lapses mentioned in the overall comments file.
Author Response
Dear Editor and reviewers,
We appreciate you for taking the time to review our manuscript entitled "Preparation, characterization and application of D-carvone-doped chitosan-gelatin bifunctional films for antioxidant and antibacterial" (Manuscript ID: foods-3762765). We have carefully revised the manuscript, and the issues raised by the editor and reviewers have been addressed as follows. All the changes are marked in red fonts throughout the revised manuscript. Page/Line numbers, Figure and Table also refer to the revised manuscript.
With Best Regards,
Hong Wu, Ph.D.
Professor
School of Food Science and Engineering
South China University of Technology
Guangzhou, China
Email: bbhwu@scut.edu.cn (H. Wu)
Responds to the editor and reviewers’ comments:
Reviewer #2: The manuscript covers an excellent and interesting topic in the field of food preservation and packaging but needs thorough revision of style, format, language corrections and in-depth analysis of some key aspects as described below section wise.
Comment 1: Title: it sounds incomplete. antioxidant and antibacterial …..?? properties, characteristics, functions???
Revision: We have revised the title and keywords based on your comments (Line 2-4).
Comment 2: Abstract: The opening sentence is not impressive and needs to be revised. For example:
In this study, composite films with dual antioxidant and antibacterial functions were prepared by combining chitosan and gelatin, with D-carvone as the primary active component. The effect of D- carvone content on the performance of the composite films was systematically investigated.
Revision: We have revised the Abstract based on your comments (Line 8-11).
Comment 3: Try to condense the methodological details and focus more on key findings (e.g., "The optimal D- carvone content of 1%-2% improved physical properties and enhanced both antioxidant and antibacterial properties").
Be specific about the performance improvements at various D-carvone concentrations (e.g., "Water contact angle increased by X% with 1% D-carvone"
Revision: We have revised the Abstract based on your comments (Line 11-19).
Comment 4: Key Words: Antioxidant and antibacterial should be replaced to improve visibility and discovery of the article in the scientific information systems as they are already there on the title.
Revision: We have revised the Abstract based on your comments (Line 27).
Comment 5: Introduction: The introduction provides a comprehensive background on the need for biodegradable and functional packaging materials. However, while the literature review is well-established, it could benefit from a more direct connection to your study’s focus. For example, briefly introduce the existing gaps in research on D-carvone-based composites at the beginning.
Revision: We have revised the MS based on your comments (Line 102-108).
Comment 6: The abbreviations must be explained on their first use. For example:
line # 65 DPPH
Line # 85-85. Repetition of “such as” in the same sentence.
Line # what do you refer by “remedicine”?
Revision: We have revised the MS based on your comments (Line 64-65, 84-86, 81-84).
Comment 7: The research gap is mentioned, but it could be clearer. Rather than stating "the majority of current research concentrates on single-function augmentation," emphasize what has not been explored in terms of dual-function (antioxidant and antibacterial) composite films specifically with D-carvone.
Revision: We have revised the MS based on your comments (Line 65-68).
Comment 8: The transition from discussing the limitations of plant extracts and essential oils to the study objective is somewhat abrupt. A smoother transition could be made by linking the limitations of previous research directly to the need for a dual-function film system using D-carvone.
Revision: We have revised the MS based on your comments (Line 68-72).
Comment 9: While D-carvone is well introduced, its role in the context of food packaging could be highlighted more explicitly. You could expand on how D-carvone’s properties make it an ideal candidate for enhancing both antioxidant and antibacterial functions, perhaps drawing from previous studies on its use in food packaging.
Revision: We have revised the MS based on your comments (Line 96-102).
Comment 10: Clarify the research gap early in the introduction to make the transition to your study’s focus smoother.
Strengthen the rationale for using D-carvone specifically in the composite film by linking its properties directly to the film’s performance in food preservation.
Revision: We have revised the MS based on your comments (Line 102-108).
Comment 11: The study objectives could be outlined more explicitly toward the end of the introduction, focusing on the dual-functional nature of your films and how this addresses the limitations in the field.
Revision: We have revised the MS based on your comments (Line 109-119).
Comment 12: Experimental:
The bacterial strains used (S. aureus and E. coli) are clearly identified, but it would be beneficial to mention their storage conditions (e.g., frozen or lyophilized) to ensure reproducibility.
Revision: We have revised the MS based on your comments (Line 125-127).
Comment 13: The preparation of chitosan (CS) and gelatin (GEL) solutions is fine, but it would be useful to provide more information on the method of dissolving these materials:
Are there any specific considerations taken to avoid clumping or incomplete dissolution of gelatin or chitosan at 60 ℃?
Revision: We have revised the MS based on your comments (Line 131-134).
Comment 14: The preparation method specifies the inclusion of D-carvone in concentrations ranging from 0% to 4%, which is good. However, the rationale for these concentrations could be discussed more thoroughly in the experimental section, especially since it was stated in the abstract that 1-2% D- carvone yields optimal properties.
Revision: Based on existing literature and preliminary experiments, we found that adding a small amount of additive significantly improves the hydrophilicity of the matrix, but adding less than 5% causes a rapid decline in the performance of the composite membrane, which requires nano-encapsulation technology to solve. We have also added explanations in the appropriate places in the MS (Line 98-102).
Comment 15: Consider adding a clarification of whether the concentrations refer to weight by weight (w/w) of the total solution or just the film-forming solution.
Revision: We have revised the MS based on your comments (Line 137-138).
Comment 16: The drying process mentions drying at 42 °C for 48 hours. It would be useful to indicate the relative humidity during drying since, which can significantly affect the final properties of the films (e.g., moisture content, mechanical properties).
Revision: We have added the appropriate content to the MS based on your comments (Line 141-143).
Comment 17: The micrometer used for measuring film thickness is mentioned, but more detail could be given on the measurement protocol (e.g., how the micrometer is calibrated or the precision of the instrument)
Revision: We have added the appropriate content to the MS based on your comments (Line 152-155).
Comment 18: How long were the films dried at 105 ℃? A time range (e.g., 6-8 hours) could be specified to avoid potential inconsistencies.
Revision: We have added the appropriate content to the MS based on your comments (Line 163-165).
Comment 19: The method for calculating solubility (equation 3) does not specify whether the films are rinsed prior to weighing after immersion. This is important, as residual water may affect the results.
Revision: We have added the appropriate content to the MS based on your comments (Line 167-169).
Comment 20: The determination of tensile strength (TS) and elongation at break (EB) is well-explained. However, it would be useful to define the cross-sectional area of the film more explicitly. Is this the area after the film is cut, or the initial dimension before cutting?
Revision: We have added the appropriate content to the MS based on your comments (Line 192-193).
Comment 21: The methodology for determining WVTR is O.K. A clarification regarding whether the sample is equilibrated in a desiccator before testing could be more appropriate.
Revision: We have added the appropriate content to the MS based on your comments (Line 198-201).
Comment 22: The SEM procedure is well described. However, it would be useful to clarify whether the gold coating is uniform and whether the thickness is standardized for all samples?
Revision: We have revised the MS based on your comments (Line 211-213).
Comment 23: The methods for measuring DPPH and ABTS+ radical scavenging capacity are described well. However, it would be helpful to mention how the control group is handled in these assays to ensure consistency.
Revision: We have revised the MS based on your comments (Line 235-238).
Comment 24: The method used for the inhibition zone assay is standard. However, specifying the medium (e.g., agar type) used for bacterial growth and the inoculation density would improve the reproducibility of the assay.
It would also be valuable to explain the reason for using UV-sterilization of the films for 30 minutes prior to testing, particularly in terms of preventing bacterial contamination during the test
Revision: We have added the appropriate content to the MS based on your comments (Line 242-247).
Comment 25: The preservation procedure is well-outlined, though further clarification regarding how often samples were monitored for quality (e.g., total colony count) during the storage period could improve reproducibility.
Revision: We have revised the MS based on your comments (Line 260-261).
Comment 26: The sensory evaluation process is detailed and appears rigorous. However, the manuscript could benefit from additional information on how subjective variations in sensory scoring were minimized? Were the evaluators trained to use a standard scale? Was the evaluation blind?
Revision: We have revised the MS based on your comments (Line 279-285).
Comment 27: The statistical analysis is clearly stated, but it may be useful to mention whether any post-hoc tests were used following ANOVA?
Additionally, including the specific test for normality (e.g., Shapiro-Wilk test) could strengthen the statistical validity of the results.
Revision: We have added the appropriate content to the MS based on your comments (Line 287-290).
Comment 28: A typographical errors in terms like "glycerol by mass of the solution" could be revised as (e.g.,“glycerol added to the solution at 2% by mass”).
Revision: We have revised the MS based on your comments (Line 134-136).
Comment 29: Results and Discussion:
As far as the discussion about the increase in thickness due to the D-carvone content is concerned, the correlation with the structural changes supported by SEM images is o.k. However,it would be useful to further discuss why a thickness increase might be beneficial or detrimental for packaging materials in practical applications (e.g., durability, flexibility, barrier properties).
Revision: We have added the appropriate content to the MS based on your comments (Line 308-312).
Comment 30: While mentioning color change due to D-carvone, it would be good to expand on how color affects consumer acceptance in food packaging? Perhaps mentioning some studies or statistics showing how consumers respond to color changes in packaging materials might enrich the discussion.
It would be interesting to discuss the potential significance of color changes, especially in relation to visual cues for freshness or quality that may affect consumer perceptions.
Revision: We have added the appropriate content to the MS based on your comments (Line 330-334).
Comment 31: The explanation of the moisture content behavior is good. However, it would be helpful to mention how moisture content might influence the shelf life and preservation of food in more detail, particularly with respect to food packaging.
Revision: We have added the appropriate content to the MS based on your comments (Line 342-345).
Comment 32: The description of the swelling rate is appropriate, but it might be interesting for the reader to know the linkage of this property to how it could affect the usability or longevity of the composite films in a real-world setting. For example, high swelling rates may not be ideal for certain applications, especially if they reduce the mechanical strength or stability of the films over time.
Could there be any practical limitations to the swelling behavior in terms of storage or packaging? A mention of how it might impact the films' ability to protect food in humid environments would enhance the value of your argument.
Revision: We have added the appropriate content to the MS based on your comments (Line 354-370).
Comment 33: Could the mechanical properties be compared to other commonly used packaging materials (e.g., polyethylene or polypropylene)? This would contextualize the data further and help to demonstrate how the composite films perform relative to existing alternatives.
Revision: Thank you to the reviewers for suggesting a comparative analysis of mechanical properties with polyethylene or polypropylene (PE/PP). Considering that this study focuses on the development of “edible, degradable” active films, PE/PP are non-edible, non-degradable materials, and their performance indicators (such as tensile strength of 20 - 30 MPa and elongation at break of 200 – 600 %) are fundamentally different from those of bio-based films. Direct comparison may obscure the environmental protection focus of this study (Line 381-386).
Comment 34: The conclusion that “too much D-carvone will lead to poor dispersion” could be better explained. It might be useful to reference studies or theories that explain how the presence of D-carvone might affect the molecular structure of chitosan and gelatin beyond just creating "poor dispersion."
Revision: We have revised the MS based on your comments (Line 381-386).
Comment 35: The analysis of WVTR is clear, and the discussion of how D-carvone affects the hydrophobicity of the films is valid. However, it would be useful to explain the practical implications of lower WVTR in food packaging. Specifically, how does the lower water vapor transmission rate impact food freshness, and does it affect respiration rates in fresh food packaging?
Revision: We have revised the MS based on your comments (Line 401-404).
Comment 36: The WCA results are well described. However, while a lower WCA suggests more hydrophilicity, the practical significance of this result could be discussed further. For example, how does the WCA affect food preservation? Is it possible for a more hydrophilic surface to lead to quicker spoilage or contamination?
Revision: We have revised the MS based on your comments (Line 415-417).
Comment 37: The discussion regarding the SEM images and the structural changes with increasing D-carvone content appears sound. The visual reference provided by the SEM images gives valuable insight into how D-carvone affects the film's surface morphology.
However, it would be useful to connect these microstructural changes to the film’s performance; e.g., Do the increased pores and roughness (above 2% D-carvone) correlate with a decline in mechanical strength or barrier properties, as discussed in earlier sections?
Revision: We have revised the MS based on your comments (Line 439-450).
Comment 38: FTIR analysis:
While the text mentions various changes in peak shapes and intensities, it would be useful to elaborate more on what these changes specifically imply in terms of molecular interactions? For instance, are these changes due to the formation of new bonds, such as hydrogen bonding, between CS, GEL, and D-carvone? This would provide readers with a more detailed understanding of the underlying chemical changes.
Revision: We have revised the MS based on your comments (Line 463-468).
Comment 39: The suggestion that D-carvone affects hydrogen bonding and cross-linking between the CS-GEL matrix is interesting, but more explanation on how D-carvone specifically disrupts these interactions would strengthen the discussion. For example, does D-carvone act as a plasticizer or reduce crystallinity? A brief mention of these mechanisms would add depth to the explanation.
Revision: We have revised the MS based on your comments (Line 446-450).
Comment 40: How does the reduced thermal stability of the composite films (due to D-carvone) affect their potential applications, especially in food packaging? It would be helpful to explain whether this reduction in thermal stability is a concern for the film’s durability during processing or storage at high temperatures.
Revision: Thanks to the reviewer for suggesting that the reduced stability of the composite material should be considered in terms of its potential applications. Considering that the TGA peak has dropped by 30 ℃, it means that the composite film may soften during the heat sealing or thermoforming process at >200 ℃, but this is far below the temperature for food sterilization (pasteurization at 85 ℃, hot filling <100 ℃) and normal storage and transportation temperatures, so it does not affect the intended applications.
The statement about the limited solubility of D-carvone and its impact on antioxidant capacity in the higher D-carvone concentration films could be clarified further. Is this issue more related to the solubility of D-carvone in the film matrix, or is it due to physical limitations in terms of D-carvone's distribution within the film?
Revision: We have revised the MS based on your comments (Line 507-511).
Comment 41: While it’s stated that the antioxidant activity may be linked to D-carvone's electron transfer capacity, it could be useful to compare the results to other natural compounds (e.g., essential oils, polyphenols) to provide context for how D-carvone compares in terms of antioxidant activity. This would help readers better understand the relative effectiveness of D-carvone in enhancing the film's antioxidant properties.
Revision: We have revised the MS based on your comments (Line 505-507).
Comment 42: The explanation of how D-carvone interacts with bacteria based on the lipophilic nature and structure of the bacteria is well-supported. However, a more detailed discussion of how D-carvone disrupts bacterial membranes or cell walls could strengthen the mechanistic understanding. Does D-carvone penetrate through lipid bilayers or interact with specific bacterial enzymes?
Revision: Thank you for your suggestion regarding the deeper significance of the antibacterial mechanism. As mentioned in the introduction regarding D-carvone, researchers have conducted extensive studies on its antibacterial mechanism, such as its ability to induce cell apoptosis, inhibit cancer cell proliferation, and exert inhibitory effects on HL-60 cell growth while inducing cell death; it can also exert its effects by inhibiting certain oncogenic pathways and oncogene expression (such as inhibiting phosphoinositide 3-kinase, a marker of cell proliferation pathways, and the expression of oncogenes such as c-fos and c-myc). Therefore, here we primarily focus on its application effectiveness in horizontal packaging.
Comment 43: It would be helpful to link these antibacterial findings to the broader context of food preservation. For instance, could these enhanced antibacterial properties result in extended shelf life for foods, and how might they compare to other preservatives?
Revision: Thank you for your suggestions regarding antibacterial properties and broader applications. Currently, this MS has only selected one representative food with high fat and moisture content for analysis. We will also consider the selection of film mechanisms based on food characteristics in the future to enable broader applications and develop its maximum application value.
Comment 44: The higher colony count observed for CS-GEL-4D after 4 days of storage is an interesting and unexpected result. It would be helpful to further investigate and discuss the possible reasons for this, particularly the effect of higher D-carvone concentrations on the film's barrier properties. Could the higher concentration of D-carvone be causing an over-activation of microbial growth due to compromised film integrity?
Revision: We have revised the MS based on your comments (Line 576-581).
Comment 45: While the antibacterial results are promising, it would be useful to discuss how long the films could maintain their antimicrobial efficacy during real-world storage conditions. What is the shelf life of food preserved with these films, and does the performance change after extended use?
Revision: We have revised the MS based on your comments (Line 581-583).
Comment 46: Conclusions: It appears that this section has been wrongly labelled as Results (Line# 535) Considering it as conclusions, following are my comments about this section.
The conclusion mentions that D-carvone content lower than 2% is optimal for film performance. However, it would be helpful to specify the trade-offs in terms of performance parameters (e.g., thermal stability, antioxidant capacity, antibacterial effectiveness) when D-carvone content exceeds 2%. What exactly happens beyond 2% that reduces the overall performance?
The conclusion could benefit from more emphasis on the practical applications of these films. How could the technology be scaled for commercial use, and what are the next steps in terms of testing these films in real-world packaging scenarios?
Revision: We have revised the Conclusions based on your comments (Line 602-619).
Reviewer 3 Report
Comments and Suggestions for Authors
Research Topic: Preparation, characterization and application of D-carvone doped chitosan-gelatin bifunctional films for antioxidant and antibacterial
Reviewer comments:
In my opinion, the manuscript entitled “Preparation, characterization and application of D-carvone doped chitosan-gelatin bifunctional films for antioxidant and antibacterial” is very interesting, and useful in a field of Food Science and Technology. However, it is necessary to revise the text further before it can be published.
Topic:
-L3-4: Should use “activities” or “properties” after "antioxidant and antibacterial.
Abstract:
-L8: Should use “activities” or “properties” after "antioxidant and antibacterial.
-L15: The full name should be used before using abbreviations.
- The experimental results should be explained more thoroughly.
Introduction:
- A more comprehensive review of research studies using D-carvone should be included.
- L94: Should change from “Pseudomonas aeruginosa” to “P. aeruginosa”
Materials and Methods:
- L121-122: Scientific names must be written in italics (Staphylococcus aureus and Escherichia coli).
- Section 2.1: Chemical reagents should specify the brand, city, and country of the manufacturer.
- L132 and L97-98: should be used consistently throughout.
- Units should be consistently formatted throughout, such as min or minute, h or hour, degree Celsius.
-L138: The full name of FTFE plate should be specified.
- L137-139: The method for controlling film thickness within the same experimental set should be explained, such as specifying the volume of solution used for film casting.
- All instruments should specify the brand, model, city, and country of the manufacturer.
- Choose either American or British English spelling consistently throughout, such as color or colour.
- "Many parameters lack proper citation methods and should be fully referenced, with detailed explanation of the analytical procedures.
Results and Discussion:
-L279-280: Should change from “p>0.05” to “p<0.05”
-L284: Should change from “FIRT” to “FTIR”
- L319: “Crataegus pinnatifida” should be written in italics.
-L295: “Table I” should be written in “Table 1”.
Conclusions:
-L535: “Results” should be written in “Conclusion”.
Figure and Table:
- Table 2 and Table 3: Units should be specified in the table headers.
References:
- Check the citation format both in the text and in the reference list.

English grammar should be checked and corrected.
Author Response
Dear Editor and reviewers,
We appreciate you for taking the time to review our manuscript entitled "Preparation, characterization and application of D-carvone-doped chitosan-gelatin bifunctional films for antioxidant and antibacterial" (Manuscript ID: foods-3762765). We have carefully revised the manuscript, and the issues raised by the editor and reviewers have been addressed as follows. All the changes are marked in red fonts throughout the revised manuscript. Page/Line numbers, Figure and Table also refer to the revised manuscript.
With Best Regards,
Hong Wu, Ph.D.
Professor
School of Food Science and Engineering
South China University of Technology
Guangzhou, China
Email: bbhwu@scut.edu.cn (H. Wu)
Responds to the editor and reviewers’ comments:
Reviewer #3:
Comment 1: Research Topic: Preparation, characterization and application of D-carvone doped chitosan gelatin bifunctional films for antioxidant and antibacterial
Topic:
-L3-4: Should use “activities” or “properties” after "antioxidant and antibacterial.
Revision: We have revised the Title based on your comments (Line 2-4).
Comment 2: Abstract:
-L8: Should use “activities” or “properties” after "antioxidant and antibacterial.
-L15: The full name should be used before using abbreviations.
- The experimental results should be explained more thoroughly.
Revision: We have revised the ABSTRACT based on your comments (Line 8-26).
Comment 3: Introduction:
A more comprehensive review of research studies using D-carvone should be included.
- L94: Should change from “Pseudomonas aeruginosa” to “P. aeruginosa”
Revision: We have revised the MS based on your comments (Line 79-119).
Comment 4: Materials and Methods:
- L121-122: Scientific names must be written in italics (Staphylococcus aureus and Escherichia coli).
Revision: We have revised the MS based on your comments (Line 125-126).
Comment 5: Section 2.1: Chemical reagents should specify the brand, city, and country of the manufacturer.
Revision: We have revised the MS based on your comments (Line 122-128)
Comment 6: - L132 and L97-98: should be used consistently throughout.
- Units should be consistently formatted throughout, such as min or minute, h or hour, degree Celsius.
Revision: We have examined the MS in the light of your comments and have made uniform changes. The modifications have been marked in red in all MS (Line 132, 140-142).
Comment 7: -L138: The full name of FTFE plate should be specified.
- L137-139: The method for controlling film thickness within the same experimental set should be explained, such as specifying the volume of solution used for film casting.
Revision: We have revised the MS based on your comments (Line 141-143).
Comment 8:- All instruments should specify the brand, model, city, and country of the manufacturer. - Choose either American or British English spelling consistently throughout, such as color or colour.
- "Many parameters lack proper citation methods and should be fully referenced, with detailed explanation of the analytical procedures.
Consistent use of American English
Throughout the text, British spellings such as “colour,” “metres,” and ‘analyse’ have been changed to American spellings such as “color,” “meters,” and “analyze,” respectively, and consistency has been maintained throughout.
Revision: Thank you for your three formatting and language requests. We have implemented each of them in the revised draft (Line 152-155, 186-187, 198-203, 211-215, 217-218, 238-240).
Comment 9: Results and Discussion:
-L279-280: Should change from “p>0.05” to “p<0.05”
-L284: Should change from “FIRT” to “FTIR”
- L319: “Crataegus pinnatifida” should be written in italics.
-L295: “Table I” should be written in “Table 1”.
Revision: We have revised the MS based on your comments (Line 299-301, 304-305).
Comment 10: Conclusions:
-L535: “Results” should be written in “Conclusion”.
Revision: We have revised the MS based on your comments (Line 602).
Comment 11: Figure and Table:
- Table 2 and Table 3: Units should be specified in the table headers.
Revision: We have revised the Tables based on your comments (Line 602, 561).
Comment 12: References:
- Check the citation format both in the text and in the reference list.
Revision: Thank you for your reminder. We have checked and standardized the full text and reference list item by item in accordance with the submission guidelines.